# 'Endlessly Valuable' Discursive Work—Intimate Partner Femicide, an English Case Study

**Adrian Howe**

Independent Researcher, Melbourne 3051, Australia; dradrianhowe@gmail.com

**Abstract:** Against the trend of roll-backs of pro-feminist initiatives by right-wing governments, feminist-led reforms to the law of murder deserve accolades as hard-fought feminist victories. For three decades, feminist analysts have critiqued the operation of provocation defences in intimate partner femicide cases. Their work has been rewarded with the implementation of reforms in several anglophone jurisdictions that have abolished or curtailed that defence. This article focuses on the revolutionary impact of the reform implemented in England and Wales. It argues for the continuing purchase for feminist legal scholars of a methodology championed by Carol Smart in her seminal 1989 text, *Feminism and the Power of Law*. She counselled feminist law scholars to read law as a site for contesting law's truth about gendered relationships. This methodology has not only been critical in exposing the misogyny and injustice embedded in traditional provocation by infidelity defences; it also enables researchers to chart shifts in law's discursive constitution of truth in the post-reform era.

**Keywords:** discourse; counter-discourse; femicide; provocation by sexual infidelity; manslaughter; murder

## 1. Introduction and Approach

At a time when, as Margaret Thornton laments, there has been a roll-back of pro-feminist initiatives by right-wing governments, this article reports on a rare 21st-century feminist legal victory, one that has flown under the radar as the *Me Too* movement dominates the headlines. For three decades, feminist analysts have critiqued the operation of provocation defences in intimate partner femicide cases. Their work has been rewarded with the implementation of reforms in several anglophone jurisdictions that have abolished or curtailed that defence. Focusing on the revolutionary impact of the reform implemented in England and Wales, I will argue for the continuing purchase for feminist legal scholars of a methodology that has been critical in exposing the misogyny and injustice embedded in traditional provocation by infidelity defences[1]. I shall suggest that, as Daniela Alaattinoğlu and I argue in *Contesting Femicide—Feminism and the Power of Law Revisited* (Howe and Alaattinoğlu 2019), a book honouring Carol Smart's seminal text (1989), her counsel to contest law's truth about gendered relationships by reading law as an important site for engagement and contestation has continuing relevance today.

## 2. Law as Discursive Site

In *Feminism and the Power of Law* Carol Smart famously advised feminists against placing all their energy in demanding law reforms, instead suggesting that we construe the legal domain as a discursive site on which to expose and challenge gendered constructions of women's experiences (1989). In short, she advocated reading law as discourse, a methodology she followed with reference to

---

[1] See Chapter One in Howe (2008) for a fuller account of Foucauldian discourse analysis.

law's constitution of the harms of rape, child abuse and pornography. *Contesting Femicide* explores law's handling of wife-killing—now 'intimate partner femicide'—a field of inquiry that was not one of Smart's focal concerns but that nevertheless provides golden opportunities for reading law cases as important discursive 'sites of engagement and counter-discourse' (Smart 1999, p. 392) and for achieving feminist law reform agendas. That intimate partner femicide cases have become critically important sites for feminist engagement and intervention across numerous jurisdictions is stunning confirmation of Smart's view that however difficult it is for feminist law reformers to achieve substantive or even symbolic victories for women, law itself remains an invaluable site for contesting hegemonic representations of gendered relationships.

First, a much-needed clarification: as we point out in *Contesting Femicvide*, Smart's warning to feminists not to get too caught up in law reform has been frequently misread as advocating a retreat from law and feminist legal activism, notwithstanding her many assurances to the contrary. Time and again Smart has reasserted her conviction that law must remain a 'site of struggle' for feminists because it provides a 'forum for articulating alternative visions and accounts' (Smart 1989, p. 88). She never wavered from the conviction that law 'understood in its widest meaning, is still one of the most important sites of engagement and counter-discourse'. The criminal justice system and legal practice remained for her important, albeit problematic—indeed 'the most problematic'—sites for feminist intervention (Smart 1999, pp. 392, 407). She has repeatedly reaffirmed this view on the continuing importance of engaging with law, most recently in an essay where she explains again that her counsel to decentre law was 'never meant to mean' that feminists should refuse to engage with law. Despite her misgivings about feminists investing too much in law reform, she was always of the view that law conceptualised as 'a site of conflict and dispute' enables feminists to 'present a constant challenge to law' and to 'intervene discursively in a high profile manner'. In short law is 'an invaluable site' for 'endlessly valuable' discursive work (Smart 2012, p. 164).

Challenging law's handling of femicide cases, feminist law scholars have approached the legal domain not only as a site on which to contest meanings about gender; they have also continued to press for law reforms aimed at curtailing victim-blaming narratives and have met with some considerable success. Nowhere is this more evident than in their largely successful challenges to the operation of defences to murder, especially provocation defences, in intimate partner femicide cases in anglophone jurisdictions. The evidence that these defences have operated in profoundly sexed ways is unequivocal—for centuries and still today, the law of provocation has facilitated men's claims that sexual infidelity provides a moral warrant for murdering 'unfaithful' wives. Feminists have been at the forefront of reform movements, for example, in England and Wales where provocation has been replaced with a new defence of loss of control. Focusing on the latter jurisdiction, I shall demonstrate how post-reform cases provide opportunities not only for assessing the effectiveness of the reforms in preventing men from getting away with murder but also for continuing the endlessly valuable discursive work that Smart practised in other fields of law. In short, they are ideal sites for continuing the work of articulating alternative accounts of gendered relationships, while challenging law's power to disqualify women's experiences of violence while privileging men's feelings and rights.

## 3. Law Reform as Discursive Site

While the provocation defence has been abolished in some Australian jurisdictions, the English law reformers chose a different path, opting for replacing provocation with a loss of control defence that expressly excludes sexual infidelity as a trigger for loss of control[2]. Leading the reform, Solicitor General Harriet Harman explained why they decided on this particular course. The provocation defence, she said, was

---

[2]　See (Tyson and Naylor 2019) for an assessment of reforms to defence to murder introduced in the Australian state of Victoria.

> . . . *our own version of honour killings* and we are going to outlaw it. We have had the debate
> and we are not going to bow to judicial protests . . . I am determined that women should
> understand that we won't brook any excuses for domestic violence . . . It is a terrible thing
> to lose a sister or a daughter, but to then have her killer blame her and say he is the victim
> of her infidelity is totally unacceptable. The relatives say 'he got away with murder' and
> they're right. (Quoted in Hinsliff 2008)

A plan to outlaw 'our own version of honour killing' struck at the heart of the received notion
that 'a crime of passion'—traditionally conceptualised as righteous slaughter in defence of honour by
a man who kills his wife or lover on finding them *in flagrante*—is a lesser crime than cold-blooded
murder. It is the excusatory force of such hot-blooded killing that has informed the provocation defence
allowing Englishmen to get away with murdering wives for centuries. Naming provocation as their
defence is an extraordinary strategy precluding the pernicious othering practice of pointing to ethnic
minority men as the typical wife-killer. No wonder it met with such virulent opposition in legal and
judicial circles.

While the English reform was informed by decades of feminist critiques of the operation of
provocation defences, it was triggered by controversial sentences handed down at the turn of the
century[3]. The *Humes* case and its contentious aftermath has been told elsewhere (Howe 2013;
Burton 2003). Suffice it to say that in December 2001, a man described by the court as an 'overworked'
solicitor, knifed his wife to death in front of their four children. He found himself 'in a red mist' and
had 'lost it totally' on learning that Maddie wanted to leave him for another man.[4] Or so he said. There
was no trial to test his evidence, the prosecution having accepted his plea of guilty to manslaughter
on the basis of provocation. He received a seven-year sentence, leaving the victim's family furious
that he was seen as the victim while 'Maddie wasn't heard at all' (Hinsliff 2003). In December 2002,
the Attorney General appealed against Humes' sentence and two others handed down in intimate
partner femicide cases on the ground of unduly leniency. In one a man had killed a woman who had
left him, the court noting he had found this 'difficult to accept'. He 'just boiled over', fell into a 'red
haze' and choked her to death. Acquitted of murder, he was sentenced to four years imprisonment for
manslaughter. The Court of Appeal declined to interfere in any of the sentences. It was unmoved even
by argument that sentences were much longer in attempted murder cases than in cases of manslaughter
by reason of provocation. As the court explained this anomaly, 'certain assumptions' had to be made
in the offender's favour in provocation cases, namely, that the loss of control was reasonable and that
the circumstances were such as to make the loss of self-control sufficiently excusable to reduce the
gravity of the defendant's offence from murder to manslaughter.[5] Being distressed by a woman who
discloses, as Maddie Humes allegedly had, that she had been unfaithful and that she wanted to leave
the marriage met this test.

With no redress in the courts on offer, it was time for legislative action. In the face of strenuous
opposition from the judiciary, the House of Lords and a wide cross section of the legal fraternity,
the reform bill abolishing provocation by infidelity as a defence to murder was finally passed. Set out
in sections 54–56 of the *Coroners and Justice Act 2009*, the new defence of loss of control came into force
in October 2010. The 2012 case of *Clinton*, conjoined appeals by three men convicted of murdering
wives who wanted to leave them, were the first post-reform femicide cases to come before the Court of
Appeal. Defying the reformers, the court unanimously determined that 'infidelity'—broadly construed
to encompass relationship breakdown—may properly be taken into consideration for the purposes of

---

[3] Incisive English analyses of intimate partner femicide provocation cases include (Allen 1988; Bandalli 1995; Edwards 1996; Burton 2003).

[4] *R v Suratan, R v Humes and R v Wilkinson* (Attorney-General's Reference No 74 of 2002, No. 95 of 2002 and No 118 of 2002) [2002] EWCA Crim 2982 at 285–86.

[5] *R v Suratan, R v Humes and R v Wilkinson* at 274. See also (Burton 2003).

the new partial defence of loss of control when such behaviour was 'integral to the facts as a whole'[6]. Had the reforms been in vain? Was infidelity to retain the excusatory force it had enjoyed in the pre-reform era?

### 4. Case Study, Intimate Partner Femicide, 2012–2016

I have discussed the controversial reform, the furore it generated and the *Clinton* case elsewhere (Howe 2012, 2013). What is of interest here is how, reading the post-*Clinton* cases as a discursive field, we can determine what has become of the traditional provocation by sexual infidelity narrative that Englishmen have resorted to for centuries. How successful has the reform banning that excusatory tale been? Are men still getting away with murder?

In *Contesting Femicide* I report on the findings of my study of men who faced trial for killing their wives, women partners and former partners in the reform jurisdiction of England and Wales in the five-year period from 2012 to 2016. I identified 317 defendants who were charged with murder from media reports, an invaluable source for tracing the reception of law's newly-reformed truth about intimate partner femicide (Howe 2019a).[7] I named the victims by way of memorialising their lives but their killers are not named, their identities being immaterial to an analysis of law's discursive construction of intimate partner femicide today. What matters is how killers and courts are putting victims, perpetrators and their defences into discourse in the post-reform age.

I will summarise the study's main findings, highlighting illuminating passages from several post-reform cases. The most significant finding was that there has been a profound shift in law's truth about femicide: defendants once characterised as impassioned killers deserving of sympathy are now being condemned as irrationally jealous murderers and sentenced to life imprisonment. Of the 240 (75 per cent of those charged) who pleaded guilty to murder or were convicted of murder by a jury convicted of murder, three received a whole life sentence, including two who had killed previous partners. The rest were given life sentences with minimum terms ranging from 11 to 38 years, far longer than sentences handed down prior to the reform. Ninety defendants admitted murder from the start or did so when it became clear during the trial that they had no viable defence. What becomes immediately clear from the guilty pleas is not only that allegations of sexual infidelity are losing their excusatory force, but that these 'red mist' cases can now be more readily identified as departure cases—where the victim was estranged from her male partner, in the process of leaving him or seeking a divorce.

The study shows that the chances of a man avoiding a murder conviction for killing a woman partner or former partner have diminished substantially. Of the 36 defendants running the new loss of control defence, 27 were found guilty of murder. In one case, that of a husband who killed his wife believing her to be having an affair, the prosecutor told the jury there must be evidence that the defendant had 'excusably lost control'. But the 'plain truth' was that any loss of self-control here was 'borne entirely of anger and jealousy at what he perceived to be his wife's sexual infidelity' (cited in Gye 2013).[8] The jury agreed. Verdict: murder. Another wife-killer went to court with the stock 'loss of control' excusatory tale, alleging she had '100 secret lovers' during their 15-year long marriage, that she had threatened to stop him seeing their children and would bankrupt if he did not agree to her terms for divorce (BBC 2015). Once the trial started, he changed his plea to murder. It took only two hours for a jury to unanimously reject the loss of control defence of a man who battered his wife to death after discovering she was having an affair. Juries also rejected defences for men who strangled former

---

6  *R v Clinton, Parker and Evans* [2012] 1 Cr App R 26 at 37.
7  The findings are drawn from a wider study of media reports of 400 cases identified from Karen Ingala Smith's 'Counting Dead Women' lists which she compiled from newspaper reports of over 680 men 'known or suspected' to have killed women in the UK over that period (Ingala Smith 2012–2016). Her lists are not confined to intimate partner femicide. For an earlier analysis of her lists see redacted 2014. Seventy-eight cases in which men committed suicide after killing partners or died in prison before their court appearance are discussed in Howe (2019b).
8  In the following, the media item reporting the trial is provided as the source for all citations.

partners for starting a new relationship; who killed them on being told their marriage was over and for one husband who struck her with force 'off-the-scale in its ferocity' during an argument about her request for a divorce (BBC 2013). One jury was unimpressed with the loss of control defence run by a man who recorded his fatal attack and his final words—'you lying cow Janee, you're dead' and 'I can't believe you're going straight from me to this guy' recorded on a voice recorder he left under a bed to spy on her. Verdict: murder. Sentencing him to life imprisonment with a 20-year minimum term, the judge explained the new legal status quo thus: one may be 'heartbroken' at a break-up, but the 'failure of a marriage and disappointment at being left for another man is relatively slight mitigation' (Arkell 2013). It could no longer found a defence.

Other defendants denied all criminal responsibility, some conjuring up increasingly far-fetched scenarios in which they had acted in self-defence. All were all convicted of murder. Combining a self-defence claim with a loss of control defence, one man said he 'lost control completely' when his partner threatened him with a knife and taunted him about his distress at the relationship ending. The jury unanimously found him guilty of murder (Mahony 2016). No defendant claiming the fatality was an accident escaped a murder conviction and once again, the context of these killings was a recent or impending separation. One estranged husband killed his wife insisted he knew nothing of his wife's affair until after her death. But the jury accepted the prosecution's case that he killed her 'out of greed, anger and possibly jealousy', being unable to accept her request for a divorce (Britten 2013). One man claimed his former partner had accidently overdosed. The prosecution case was that he had harboured thoughts of a reconciliation but killed her when he saw she had removed all his photos. The judge explained law's new understanding of relationship breakdown's depleted excusatory force: feeling 'hurt and puzzled' when your wife asks for a separation or removes your photos is no excuse for murder. Wanting to 'move on with her life' even if he 'harboured thoughts that there would be some sort of reconciliation' was something a woman was 'entitled to do' (McCarthy 2014). In the post-reform cases, her newly recognised entitlement is cancelling out his traditional entitlement to a conviction for manslaughter, all of these 'accidental' killers receiving life sentences for murder with minimum terms ranging between 22 and 25 years.

The last woman to be killed in the five-year period covered by my study was stabbed 20 times by her partner on Christmas Day, 2016. Repeated denials of her supposed infidelity had enraged him. 'People just snap, you know what I mean', he told the police (BBC 2017a). With the historically-mandated 'snapping' defence now banned as a trigger for loss of control, he pleaded guilty to murder and was sentenced to life imprisonment with a minimum sentence of 23 years. Court dispositions in post-femicide cases suggest that defendants who were once characterised as impassioned killers deserving of sympathy now tend to be condemned as irrationally jealous murderers. In one departure case, the prosecutor told the jury that sexual jealously 'can't excuse an unlawful killing. However humiliated and upset he felt, there was no justification for killing her in broad daylight, and we say this is murder' (Murdermap 2013). Nor did pleading guilty to murder save another defendant from a long sentence: life with a minimum term of 20 years. The judge regarded the 'considerable emotional strain' he was under as 'significant' but being 'understandably upset' about the relationship ending did not warrant stabbing her 13 times. After all:

> Many men and women have to endure the discovery that the husband, wife or partner is no longer content with the relationship they have … Many are jealous or unhappy. But what the law cannot and will not permit is the use of violence. (Osuh 2012)

If the number of convictions and long sentences for murder are any guide, a victim's 'innocence' or, more commonly, lack thereof—central to the traditional provocation defence—appears to be losing its currency with the growing recognition of a woman's right to leave a relationship notwithstanding the stress it might cause her partner. But not always. Juries sometimes accepted loss of control defences, resulting in convictions for manslaughter and sentences ranging from as low as six years, reminiscent of the pre-reform era. Amongst other troubling cases, a jury sympathised with an estranged husband

who killed her when she made the 'bombshell' revelation that he was not the father of their eldest son. He said she made the revelation during an argument when she arrived with papers relating to the sale of the house, taunting him with 'You aren't half the man you think you are' and the new man was 'a lot better man than you' (Robinson 2015). Another jury unanimously accepted a man's account of losing control when his wife allegedly threatened him with a knife as their relationship deteriorated. The judge's pronouncement—'People who wish to end relationships and live alone without their former partner are entitled to do that without being killed for it'—suggested a longer sentence than the seven and a half years imposed for manslaughter (BBC 2014a). In another telling case, a defendant said he 'snapped' when his wife admitted having sex with a lover and was planning to leave him. The jury was told that legally, his claim that his wife had admitted having sex with another man was not a defence to murder. But if they believed she had threatened to leave and that was the trigger for him to snap, they could acquit him of murder. Might someone, the defence asked, who 'realised both the deception he has been subjected to and the fact his life was about to be utterly shattered, snap, as he did?' After three days of deliberation, the jury agreed—killing an unfaithful woman who 'threatened' to leave was not murder (Calderwood 2016).

　　　As for victims' families, one was outraged that the jury had cleared the defendant of murder. The case was reported under the heading: 'Manslaughter Injustice'. Aware that the killer would serve only half of his 18-year term, her son despaired: 'Nine years for a man that's capable of killing someone and burying them on the moors—there's no justice' (Rush 2013). Another family also felt let down by the jury. While the prosecution called the stabbing by a woman's former partner a 'brutal and sustained attack' in revenge for rejecting him, the defence claimed she had taunted him about a previous partner and said some 'horrible nasty vindictive stuff' to him causing him to lose control. Family members were reported to have rushed distraught from the court when the jury foreman announced a unanimous verdict of not guilty to murder after two days' deliberation. An incredulous family friend was 'disgusted' at the outcome. She was 'not expecting that at all'. The family thanked the judge 'for coming to the only sentence suitable for this crime'—life with a minimum term of seven and a half years, but not to be released until he was considered he was no longer a danger to the public (read: women). As for the family friend, she remained perplexed: 'I'm just wondering, what is murder?' (Passant 2015).

　　　Most juries are not left wondering. They are falling in line with law's new truth that, even taking into account on evidence most favourable to the accused, notably testimony about the victim's conduct leading to loss of control, most wife-killing is murder. However, residual resistance to depriving distraught wife-killers of a defence can be found even in a case where the defendant admitted murder. Having stabbed his wife 51 times in front of a witness whom he also attacked, one defendant pleaded guilty to murder. His lawyer then convinced him to apply to withdraw his guilty plea, indicating—as the prosecution observed—'a clear dichotomy' between the defendant's and his lawyer's accounts of the fatal attack. The application was rejected. At the sentencing hearing, the prosecution argued that there was no loss of control over his wife's affair which he had known about for some weeks. Instead, the attack marked the end of weeks of 'intolerable intrusion' into his wife's life, a 'campaign of surveillance' no less. It was a murder 'committed out of jealousy, pride and anger' over his wife's announcement the marriage was over. According to the defence, 'the reality' was 'more nuanced' than that and the stress caused by the deteriorating marriage and his wife's affair was surely a mitigating factor. The judge agreed, taking three years from the sentence in mitigation (Humphries 2015). Why being stressed in such circumstances should mitigate at all is surely inconsistent with the judicial pronouncement cited earlier that a woman is entitled to leave a relationship 'without being killed for it'. Nevertheless, a sentence of life with a minimum term of 26 years was nearly four times the sentence imposed for manslaughter in *Humes*. For the prosecutor, the judge and even the defendant initially, her killing was clearly murder. Only the defence lawyer thought otherwise. Having failed to show it could be defended, all that remained was a plea in mitigation for a client light years removed from the easily defended impassioned wife-killer of old.

## 5. Diminished Responsibility in Intimate Partner Femicide Cases, 2012–2016

My 2012–2016 study also indicated that with provocation by sexual infidelity off the table as a defence, some defendants opted for a diminished responsibility defence, almost all unsuccessfully. The new diminished responsibility plea introduced by Section 52 of the *Coroners and Justice Act 2009* provides that a person not be convicted of murder if suffering from an abnormality of mental functioning that arises from a recognised medical condition and that substantially reduced their responsibility for their actions. This new wording appears to give more scope to the importance of expert psychiatric evidence in determining criminal responsibility but the jury remains the ultimate arbiter (Reed and Bohlander 2011). How then are juries responding to diminished responsibility in the post-reform era?

I identified 7 diminished responsibility cases over the five-year period of my study (Howe 2019b, pp. 60–65). Prosecutors accepted 41 pleas of guilty to manslaughter, almost all from men suffering from psychosis, dementia or delusional disorders. If the manslaughter plea is rejected the case proceeds to court. Unless there is clear evidence of mental impairment, these defendants are usually convicted of murder. In 30 of the 34 cases that went to court juries found them guilty of murder. Over half were departure cases, juries evidently unimpressed with psychiatric testimony about medical conditions that diminished the defendants' criminal responsibility. Lengthy minimum sentences for murder ranging from 20 to 28 years indicate judge and jury were on the same page. One defendant had threatened to 'slice her open' when his former partner ended their relationship. He stabbed her to death when she arrived to collect her belongings from their flat, later telling the jury: 'I wanted to hurt her, just to make her feel how I felt'. A consultant forensic psychiatrist said he had an 'adjustment disorder', described as 'a reaction to a stressful life event, such as a bereavement or a separation' which 'fit in between more severe mental illness and 'a *normal reaction*'. It could, he said, cause 'quite dramatic behaviour' (McQueeney 2012; my emphasis).' They jury rejected such fine distinctions, deciding the dramatic behaviour was murder. That's the new normal in post-reform cases.

The *BBC News* reported another femicide under the matter-of-fact headline 'Christopher Parry shot wife Caroline dead "because she left"' (BBC 2014b). He said he only meant to shoot himself. Psychiatrists disagreed about his mental health. Two said he was severely depressed and this had substantially impaired his responsibility, while a third concluded his depressive illness before the shooting was 'mild-to-moderate'. The jury preferred the prosecution account that the shooting was a carefully planned act of murder by a man 'not prepared to let go' (BBC 2014c).

In another revealing case, psychiatrists disagreed about whether a defendant was mentally ill when he stabbed his partner to death, almost decapitating her. She had also been strangled and hit with an iron. In total there were over 60 injuries, many thought to have been inflicted after death. He was finally found fit to plead after a year-long investigation into his mental state. A witness testified he heard him tell the victim: 'If you will leave me, I shall kill you' and 'if I go to prison, I will pretend I'm mad. They will put me in hospital and I will get free from there'. The prosecution's case was that he killed her because he was enraged at the prospect of her finally leaving him and it was no coincidence that she was 'packing her suitcase' when he killed her. It was 'a killing committed in anger and jealousy'. The jury agreed. Sentencing him to life with a 20-year minimum, the judge said the murder was 'the product of narcissistic rage' at her departure and could 'properly be described as the slaughter of an innocent woman' (BBC 2017b)—a victim's innocence remaining a factor to be noted during sentencing. Still, times are changing. One defendant who stabbed and bludgeoned his partner to death gave 'little indication as to why he killed her', leaving police to speculate that 'he may have become aware that she was planning on ending their relationship' (Cooper 2016). That, after all, is the predominant theme in intimate partner femicides and diminished responsibility cases are no exception. The killing of a departing woman, whether attributed to a loss of self-control or mental impairment of some kind, is now being registered as murder in England and Wales.

What though of exit cases complicated by a man learning of his wife's affair around the same time he hears she is leaving him? One defendant said he stabbed her to death a day after he learnt that she was planning to live with the new man and hearing her say: 'I love him more than you' (Osuh 2017a).

Prior to the reforms tightening up partial defences to murder, this was the kind of killing scenario that led to successful provocation defences resulting in convictions for manslaughter and sentences ranging between four and seven years. Not now. This jury unanimously found him guilty of murder. As for killings committed in a non-departure context, one defendant killed his partner over unfounded suspicions she had been having sex with another man. A psychiatrist testified that he was an alcoholic suffering from depression, anxiety, agoraphobia, amnesia, suicidal thoughts and fears that he would be sectioned. But in the jury's view, his savage attack was murder (Derbyshire Times 2016).

The high failure rate for mental impairment defences indicates that most juries today have low tolerance for excuses offered by overwrought men who kill 'their' women. That 30 of 34 diminished responsibility defences failed suggests that the new tighter defence—one not part of the feminist law reform agenda—is operating even more effectively than the new loss of control defence that they advocated. Arguably though, the feminist challenge to the partial defence of provocation has had a flow-on effect in all intimate partner femicide cases. Read as discursive fields, they show that both reformed defences are achieving the feminist goal of preventing men getting away with murder, not least by exposing the 'unfaithful wife' narrative as code for 'bereavement separation' from a departing one. For the most part, juries are rejecting tales of woe said to be precipitated by an unfaithful or exiting woman. However, some non-coping killers are evading murder convictions, notably three who drew on the all too familiar divorce narrative to plead not guilty of murder on the grounds of diminished responsibility due to separation-induced depression. One had become depressed when he sensed his wife's attitude towards him had changed. Was she cheating on him? Five days before he killed her, she asked for a divorce. So, when she insulted him during an argument, he showed her a knife, ostensibly to 'quieten' her, but ended up stabbing her to death. His lawyer likened the offence to 'a crime of passion', the very crime the reformed law of murder was designed to curtail. After hearing testimony from a psychiatrist that he would not have committed the offence but for an 'adjustment disorder', the jury cleared him of murder. The judge noted that while 'a human life has been lost', the marriage had been 'happy' and the defendant 'a good, placid and kind husband' who had been greatly distressed by changes in his life (Leicester Mercury 2013). He imposed a four-year sentence reminiscent of the risibly short sentences that had helped precipitate the reform movement. In a classic throwback to the taunting cases of old, another defendant with a depressive illness said he killed his wife because she had goaded him about his small penis and alcohol-induced bed-wetting and tried to kill himself. She had wanted to leave him because of his excessive drinking. Tellingly, blood at the scene was daubed over papers marked 'ending a marriage' (Stewart 2013). The jury took less than three hours to find him not guilty of murder, testament to compassion's longevity in some quarters for men distressed by a departing, verbose wife.

## 6. Law's Truth Today

Today most juries and some judges recognise that her right to leave trumps his right to a conviction for manslaughter. Law's authoritative discourse is starting to register a new truth about the 21st-century English wife-killer and his victim. While exculpatory judgments about placid and kind husbands who just 'snap' when hurt still occur, the impassioned wife-killer of old is transforming in legal discourse into a dangerous offender, a pathetic, needy one at that, his credibility finally starting to falter in the face of feminist challenges to law's long-standing truth that killing 'your' woman is a lesser form of homicide than murder. Indeed, the evidence unearthed in the study suggests that courts have heeded the new legal status quo captured in the 2003 newspaper headline, '"Crime of passion" is no defence', heralding the planned shake up of the law of murder (Hinsliff 2003).

Analysis of court outcomes for intimate partner killings committed after the period covered by my study reveals the same pattern of failed defence narratives[9]. Kulwinder Kaur's husband stabbed her to death in January 2017 after discovering her and a man in the family home. He claimed he had lost his self-control after his wife had taunted him about his sexual prowess. The jury convicted him of murder. In a case reported by the BBC under the headline 'David Clark jailed for killing wife over penis taunts', the defendant claimed he had lost control when he killed his wife Melanie in December 2017. He said she had belittled the size of his genitals, informed him of a lesbian affair and made other cruel remarks (BBC 2018).[10] The jury convicted him of murder. Sentence: life with a 15-year minimum, a notable increase over sentences handed down in 'penis-taunting' cases in the pre-reform era. The successful appeal against a conviction for murder in a pre-reform era penis-shaming case that I have discussed elsewhere is worth recalling here (Howe 2004). In the 2003 case of *Rowland*,[11] the applicant wife-killer argued that fresh psychiatric evidence should be heard about the relationship of his depression to the provocation he had endured from his wife. She had, amongst other things, a drinking problem, she had withdrawn sexually, made him feel jealous and taunted him about the bend in his penis caused by scar tissue from a medical condition he suffered from. On the fatal night, she refused to engage in a conversation with him about their relationship and continued to taunt him about his bent penis, causing him to lose his temper and knife her to death. Three appeal judges quashed his murder conviction and sentence of life imprisonment, substituting a verdict of manslaughter by reason of provocation and a seven-year sentence. The sharply different outcomes in these pre-reform and post-reform 'penis taunting' cases encapsulate how impactful the feminist-led law reforms to defences to murder have been.

With defences to murder now tightened up, many defendants are left with no option but to enter pleas of guilty to murder. They included Chrissy Kendall's estranged husband who police said he had taken their split 'badly' and strangled her during an argument in April 2017. According to his defence lawyer, she had told the defendant that:

> . . . enough was enough, and that while they had cohabited while estranged, she was now leaving. It was at that point the defendant saw red. It was at that point the defendant lost all sense of reason. This was a moment of madness, when he was gripped by an inability to accept the breakdown of his family unit, the loss of the woman he loved and still loves'. (Osuh 2017b)

With 'seeing red' over a wife's departure no longer founding a defence, he pleaded guilty to murder. So did Karolina Chwiluk's husband. The judge said the fatal attack was 'motivated by jealousy and the fact she had made clear the relationship was over' (Press Association 2017). He received the mandated life sentence with the minimum sentence set at 20 years. Sentencing Linda Parker's ex-partner to life with a minimum of 22 years after he admitted killing her, the judge said:

> The facts of this case illustrate the tragedy of the loss of a life, set against a sadly all too familiar backdrop of a history of domestic violence, where one partner finally manages to end the relationship, only for their ex-partner to be unwilling, or unable, to accept that the relationship is at an end leading to a confrontation and fatal injuries to the victim, for that is what Linda is, a victim of your criminal conduct.
>
> (Sentencing Remarks 2018a, *R v Glen Gibbons* para.21)

---

9   Here I will revert to my practice of naming the victims. The victims in my 2012–2016 study are all named in (Howe 2019a, 2019b).

10  Media coverage of this case was criticised as 'salacious' and demeaning to the victim (Starling 2018).

11  *R v Rowland* [2003] EWCA Crim 3636, Dec 2003.

This gender-neutral framing of the problem of escalating 'domestic' violence notwithstanding, the judge's response to the defendant feeling 'rejected and abandoned' and unable to 'deal with the ending' of their relationship bears recording:

> To the extent that you were in love with Linda it was an unhealthy and controlling love, and one that did not prevent you being violent to her during your relationship, and which ultimately led to her untimely death as a result of your actions.
>
> (Sentencing Remarks 2018a, *R v Glen Gibbons* para.30)

Such 'unhealthy and controlling love' is now leading to incontestable murder convictions for rejected men who follow through with fatal violence against their partners. Susan Gyde's husband was another who pleaded guilty to murder. After killing her he sent her lover a text message containing a variation on the standard killing script of old: 'If I can't have her, you can't either. She's dead' (Vernalls 2019), thereby sealing his fate in the post-reform age. Cristina Magda-Calancea's boyfriend did not have a viable defence, his tale about her having 'romantic relations' with another man leading him, in a 'moment of passion', to lose control and stab her no longer having any purchase in the courts (Walsh 2019).

Failed diminished responsibility defences continue to highlight law's new truth about intimate partner femicide: no matter how upset or psychologically challenged these killers are, their crimes are murder, not manslaughter. Karina Batista's husband killed her in February 2017 after wrongly accusing her of having an affair. There was 'some evidence of mental disorder' that had 'some minor role in the commission of the offence' but clearly it was not enough for the jury. Verdict: murder. Sentence: life with a 20-year minimum (Minchin 2018). Molly McLaren's ex-partner tried a diminished responsibility defence after stabbing her more than 75 times two weeks after she ended their relationship in June 2017. The jury took less than four hours to convict him of murder. The judge said he might never be released: 'You were determined to punish her for ending the relationship with you. You were seeking revenge' (Siddique 2018).

Revealing too was the fate of Tyler Denton's killer. Convicted of her murder and the attempted murder of three members of her family in September 2017, he was sentenced to life with a 30-year minimum, despite having a defence psychiatrist testify that he suffered from a mental disorder which significantly impaired his ability to control himself. The psychiatrist felt his 'schizotypal disorder' had been 'present for a number of years, with the defendant displaying seven of the 14 possible symptoms' such as feelings of frustration and sadness. Having hoped for a longer-term relationship with his victim, he 'felt frustration and sadness, but he specifically denied anger' when it ended (Roberts 2018). The jury was unmoved as it was in the trial of Saeeda Hussain's husband. He killed her in February 2018 having wrongly suspected she was having an affair. The judge said his 'mounting paranoia and abuse of his wife was evidence he was suffering from "Othello syndrome"' and that there was 'no evidence to suggest that relationship for the first 50% of the marriage was anything other than a satisfactory and an amicable one' (Rodgers 2019). The jury must have focused more on the second 50% of the marriage where the defendant's coercive control and escalating violence culminated in an attack featuring at least 46 blows from a machete and seven from a hammer than on any syndrome evidence. Verdict: murder. Sentence: life with a 20-year minimum.

## 7. A History of the Present

As I write, Ann Marie Pomfret's husband has been convicted of murdering her in November 2018. Initially he denied any involvement in her death but when the evidence against him became overwhelming, he admitted manslaughter due to loss of control. He claimed to have lost self-control when his wife taunted him, calling him 'limp', which he believed referred to his problems with erectile dysfunction. So, he hit her over the head with a crowbar more than 30 times. His barrister described him as a 'man of impeccable character', who for many years, had been 'a model of self-control, patience and restraint', a 'quiet man who finally snapped', having suffered verbal and physical abuse at her

hands (Crook et al. 2019). Verdict: murder. Sentence: life with 20-year minimum. A defence narrative focused on the victim's alleged taunts that might have carried weight in the pre-reform period no longer had a purchase.

Just a few months earlier, Barbara Davison's partner pleaded guilty to murdering her and was sentenced to life with a 23-year minimum. At first, he claimed he did not mean to kill Barbara. He only meant to 'silence' her during an argument by smothering her. He changed his plea shortly before his trial. It emerged that he had been convicted of manslaughter for killing a previous partner, Jacqueline Aspery, in 1996. He was given a three-year sentence. After his sentence was handed down for the second killing, Jacqueline Aspery's daughter Lorna Peters vented her rage at the sentence he had received for killing her mother:

> I challenged that judge to come down and sit down with me and explain why he only got three years. The whole trial was a farce. I didn't get justice for my mum. I was absolutely disgusted. (Jones 2019)

> The prosecutor at his second trial put that point this way: the jail time he had received for his earlier killing was 'profoundly merciful by today's standards'. (Jones 2019)

When leading the movement to reform the English law of murder, Solicitor General Harman was said to have favoured abolishing provocation outright (Hinsliff 2003)[12]. Might that have been more effective in curtailing residual sympathy for men killing unfaithful or departing women? While conclusive findings must await a more extensive study of pre-and post-reform cases, I will conclude with a preliminary assessment of the English reformers' claim that they were implementing 'an important law change that will end the culture of excuses'; that changing it would 'end the injustice of women being killed by their husband then being blamed' and that 'the days of sexual jealousy as a defence are over' (Scoop Independent News 2008).

Are those days over? Not quite. Loss of control defences can still succeed. In the post-reform period, a few defendants were given sentences for manslaughter of between four to seven years, on a par with the pre-reform sentences that had so outraged families and reformers. Just as concerning was the judge's direction that the jury could accept a loss of control defence if they believed her 'threat' to leave was the trigger making him snap. By reinforcing the pre-reform notion that a man provoked to kill a departing woman is guilty of manslaughter only, such a direction highlights the limitations of a reform confined to excluding sexual infidelity as trigger for loss of control. Yet while residual jury sympathy for some enraged killers and occasional risibly short sentences remain causes for concern, the English reform appears to be working to curtail both unfaithfulness and departure as excuses for murder. So many guilty pleas and convictions for murder are testimony to that. The historically mandated provocation by infidelity script followed by so many men in so-called 'crimes of passion' is being exposed as code for a man's murderous rage at being abandoned by a woman who, as criminal courts are increasingly recognising, is entitled to leave a relationship without fatal consequence. Law's traditional wife-killer, reproduced over the centuries as an impassioned man provoked to kill an unruly and unfaithful wife in a spur-of-the-moment 'red mist' frenzy, is a shadow of his former self. That legendary *criminel passionel* emerges from the post-reform cases discursively depleted, a man incapable of coping with relationship breakdown who kills 'his' woman in a frequently planned, far from spontaneous attack, only to find that alleging loss of control in the face of such adversity no longer has much excusatory force. The reform is having its intended effect—men, at least most of them, are no longer 'getting away with it'.

---

12　Australian feminists have long advocated abolishing provocation outright, that is both as a defence and as mitigation at sentencing. See for example (Howe 1998, 2002, 2004; Tyson 2013).

### 8. Conclusions

When Carol Smart warned in *Feminism and the Power of Law* that feminists can be too easily 'seduced by law', that they tended to cling to the naive hope 'that new law or more law might be better than the old law', that indeed, in exercising law, they may 'produce effects that make conditions worse' for women, she was referring specifically to reforms that extended law into 'new terrains' (Smart 1989, pp. 160–62). Reforms to defences to murder do no such thing. They actually work in the way that Smart advocated, de-centring and retracting law, disqualifying its age-old truths about the excusatory force of men's emotional turmoil. Far from deploying law 'in the cause of women', or extending 'law's imperialist reach', these reforms have been an exercise in withdrawing law from the cause of men (Smart 1992, p. 30). It has been a mostly successful challenge to law's power to define 'the truth' of intimate partner femicide, a challenge furthermore that has read law as Smart counselled, as a site of discursive struggle and as a forum for articulating an alternative feminist account of gendered relationships, this time of their termination in 'infidelity' femicide (Smart 1989, pp. 88, 164).

This has been a rare feminist victory. Not that the battle is over. Provocation can still be taken into account at sentencing, allowing victim-blaming to continue there. For example, in the 2017 'penis taunts' case, the judge took into account the 'mitigating factor' of provocative conduct in the hour leading to her murder—her taunting about 'the claimed sexual encounter' with a woman and a message that 'everyone had been laughing' at the defendant for years (Sentencing Remarks 2018b: *R v David Clark* 2018, p. 5). And while victim-blaming narratives are rarely successful today, they still get a hearing during the trial. Sheila Thomas left her husband for a former lover who had traced her via Facebook. Her husband demanded she leave the family home then killed her when she returned home to collect belongings in July 2017. He testified that she had said: 'I don't love you … I have never loved you'; that she 'detested' having sex with him, that she should have left him years ago and that he was not the father to two of their daughters. (O'Brien 2019). The jury found him guilty of murder by a majority of 11 to one but it took over nine hours to debate the impact of such alleged verbal abuse on a man's self-control.

Finally, the May 2017 Tracy Kearns case demonstrates that the traditional 'sexual infidelity' narrative, supposedly banned by legislative reform, can still on occasion save a killer from a murder conviction. Her partner told the court he had lost control and killed her because she had compared him unfavourably to her lover. The jury accepted his defence. The victim's family was 'devastated' by the manslaughter verdict and the 13-year sentence, her mother declaring:

> I will never understand how this can be seen as justice. Nothing other than a life-sentence would ever be justice for what this man did to Tracy. (ITV Report 2017)

Justice for intimate partner femicide victims in the form of a murder conviction is what families want and reformers promised, and that is what most of them are getting in the post-reform era. Law's truth about these victims and these killers has undergone a long overdue though rapid transformation. There is then plenty of work ahead for a new generation of feminist legal scholars. They would do well to continue exploring post-reform cases—perhaps especially the problematic ones—as a rich supply of discursive sites for that endlessly valuable feminist work of contesting law's truth.

**Funding:** This research received no external funding.

**Conflicts of Interest:** The author declares no conflict of interest.

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
