# Peer review of "‘Endlessly Valuable’ Discursive Work—Intimate Partner Femicide, an English Case Study"

_laws, 2019_

Round 1

Reviewer 1 Report

This was a terrific paper. I felt that the introduction/abstract could use some editing. The first sentence was too long and difficult to unpack: "Reforms to the law of murder informed by three decades of feminist critique of the operation of provocation defences in intimate partner femicide cases have led to the abolition or curtailment of that defence in several anglophone jurisdictions." It could use with breaking up the sentence and putting in plain language. I found that I did not understand what the article was about from the abstract and I was quite a ways into the paper before I understood that meaning of the opening sentence in the abstract. Similarly, this sentence is overly dense and does not convey the meaning of the article: " It argues for the continuing purchase for feminist legal scholars of the foucauldian-inflected discourse analysis."

While the abstract was overly-dense, I found the substance of the article extremely accessible and easy to follow. The author(s) made excellent use of stories and cases throughout to illustrate and support their argument. One thing I would have liked to have seen is the number of cases brought each year on average and the overall sentencing results just to give a sense of the scale of these cases. That may be beyond the scope of this project, however. 

This was a strong paper and a worthy contribution to the literature on intimate partner violence and legal process.

Author Response

Review 1

This was a terrific paper. I felt that the introduction/abstract could use some editing. The first sentence was too long and difficult to unpack: "Reforms to the law of murder informed by three decades of feminist critique of the operation of provocation defences in intimate partner femicide cases have led to the abolition or curtailment of that defence in several anglophone jurisdictions." It could use with breaking up the sentence and putting in plain language. I found that I did not understand what the article was about from the abstract and I was quite a ways into the paper before I understood that meaning of the opening sentence in the abstract. Similarly, this sentence is overly dense and does not convey the meaning of the article: " It argues for the continuing purchase for feminist legal scholars of the foucauldian-inflected discourse analysis."

While the abstract was overly-dense, I found the substance of the article extremely accessible and easy to follow. The author(s) made excellent use of stories and cases throughout to illustrate and support their argument. One thing I would have liked to have seen is the number of cases brought each year on average and the overall sentencing results just to give a sense of the scale of these cases. That may be beyond the scope of this project, however. 

This was a strong paper and a worthy contribution to the literature on intimate partner violence and legal process.

Response:  Abstract and introduction amended.

Providing yearly averages and overall sentencing results is indeed, as the reviewer surmises, beyond the scope of this paper.

Reviewer 2 Report

Overall, this paper was generally well written and as a feminist scholar, filled me with hope! The most significant changes I would like to see relate to the author’s engagement (lack thereof) with the Australian research literature in this area. There are also some typographical errors and clumsy wording that should be addressed. Specific comments appear below.

Typo “reaffirming” should be reaffirmed second page The author states the following – “Feminists have been at the forefront of reform movements to abolish provocation, most successfully in several Australian jurisdictions which have banned the defence outright.” Then again it is noted that, “While abolition of the provocation defence has been the reform favoured in several Australian jurisdictions”. These statements are misleading. In Queensland provocation can be used as a full defence to assault charges and as a partial defence to a murder charge. See sections 268, 269 and 304 of the Criminal Code Act. Feminist socio-legal scholar continue to fight for reform in this jurisdiction. In NSW in response to feminist concerns the provocation was replaced by a partial defence of “extreme provocation.” However, ‘extreme provocation’ has been critiqued by feminist socio-legal scholars for not going far enough (see Fitz-Gibbon, Kate --- "Homicide Law Reform in New South Wales: Examining the Merits of the Partial Defence of 'Extreme' Provocation" Melbourne University Law Review 40, 3). In Victoria, provocation has been abolished but replaced by defensive homicide. Feminist socio-legal scholars have argued that in the absence of provocation, defensive homicide has simply become a replacement excuse for men’s violence against women (see Bronwyn Naylor, Danielle Tyson: Reforming Defences to Homicide in Victoria, International Journal for crime, justice and social democracy, 6(3): 72‐87). Western Australia still has a provocation defence (see Criminal Code Act Compilation Act section 246) The Northern Territory still has a provocation defence (see section 158 of the Northern Territory Criminal Code). In South Australia, the State Government introduced legislation abolishing the defence of provocation but this won’t come into effect until the end of 2019. The Australian Capital Territory still has a provocation defence (see section 13 of the Crimes Act) Tasmania has abolished the provocation defence. Typo in the following statement…..As for victims’ families, one was outraged that the jury had clearing the defendant of murder”. “62 groups of injuries” should this read 62 separate injuries? “As for killings committed in a non-departure context, one defendant killed his partner killed over unfounded suspicions she had been having sex with another man. A psychiatrist testified that he was an alcoholic suffering from depression, anxiety, agoraphobia, suicidal thoughts and fears he would be sectioned and amnesia. But in the jury’s view, view his savage attack was murder (Derbyshire Times 2016). Clumsy/unclear wording. Needs to be re-written. “The defendant claimed he had lost control when his killed his wife Melanie in December 2017”. Typo. “his wife his wife”. In fact, the purpose of this paragraph needs to be made clear. I am assuming that the point in outlining the 2003 and 2017 cases is to show/compare how the courts approach to penis taunting has change post reform? This needs to be made clear to the reader.   “A row” very British term…replace with an argument. “Sentence: lie with a 20-year minimum. He received the mandated life sentence with the minimum sentence set at 20 years”. Confusing, delete “lie with a 20-year minimum”. Again, referring to the Australian situation the author notes, “when leading the movement to reform the English law of murder, Solicitor General Harman was said to have favoured abolishing provocation outright, the preferred option in Australian jurisdictions”. Arguably, as per my previous comments, this is not the preferred option in Australia. The author then goes on to note that, “while conclusive findings must await more extensive comparative studies of pre-and post-reform cases across reform jurisdictions” (i.e. in Australia vs the UK). I would like to direct the author to the work of Kate Fitz-Gibbon in Australia who has been doing these types of analysis see https://www.researchgate.net/profile/Kate_Fitz-Gibbon. Any discussion regarding the Australian context should be made in references to Kate’s work as she is the leading authority on this subject. Overall, I would argue that the author should engage more directly with the Australian literature in the area and draw out points of comparison. This would strength the paper significantly.

Author Response

Review 2

Overall, this paper was generally well written and as a feminist scholar, filled me with hope! The most significant changes I would like to see relate to the author’s engagement (lack thereof) with the Australian research literature in this area. There are also some typographical errors and clumsy wording that should be addressed. Specific comments appear below.

Typo “reaffirming” should be reaffirmed second page.  

Amended.

The author states the following – “Feminists have been at the forefront of reform movements to abolish provocation, most successfully in several Australian jurisdictions which have banned the defence outright.” Then again it is noted that, “While abolition of the provocation defence has been the reform favoured in several Australian jurisdictions”. These statements are misleading. In Queensland provocation can be used as a full defence to assault charges and as a partial defence to a murder charge. See sections 268, 269 and 304 of the Criminal Code Act. Feminist socio-legal scholar continue to fight for reform in this jurisdiction. In NSW in response to feminist concerns the provocation was replaced by a partial defence of “extreme provocation.” However, ‘extreme provocation’ has been critiqued by feminist socio-legal scholars for not going far enough (see Fitz-Gibbon, Kate --- "Homicide Law Reform in New South Wales: Examining the Merits of the Partial Defence of 'Extreme' Provocation" Melbourne University Law Review 40, 3). In Victoria, provocation has been abolished but replaced by defensive homicide. Feminist socio-legal scholars have argued that in the absence of provocation, defensive homicide has simply become a replacement excuse for men’s violence against women (see Bronwyn Naylor, Danielle Tyson: Reforming Defences to Homicide in Victoria, International Journal for crime, justice and social democracy, 6(3): 72‐87). Western Australia still has a provocation defence (see Criminal Code Act Compilation Act section 246) The Northern Territory still has a provocation defence (see section 158 of the Northern Territory Criminal Code). In South Australia, the State Government introduced legislation abolishing the defence of provocation but this won’t come into effect until the end of 2019. The Australian Capital Territory still has a provocation defence (see section 13 of the Crimes Act) Tasmania has abolished the provocation defence.

Response: My article is about the continuing purchase of an approach to the law question that views law as a site for contesting law’s truth, using English not Australian cases and reforms as a case study. It is not a comparative study.

I have amended the sentence said to be misleading about Australian reforms. I have also amended the title to make it clear this is an English case study.

Typo in the following statement…..As for victims’ families, one was outraged that the jury had clearing the defendant of murder”. “62 groups of injuries” should this read 62 separate injuries?

Amended.

“As for killings committed in a non-departure context, one defendant killed his partner killed over unfounded suspicions she had been having sex with another man. A psychiatrist testified that he was an alcoholic suffering from depression, anxiety, agoraphobia, suicidal thoughts and fears he would be sectioned and amnesia. But in the jury’s view, view his savage attack was murder (Derbyshire Times 2016). Clumsy/unclear wording. Needs to be re-written.

Amended.   

“The defendant claimed he had lost control when his killed his wife Melanie in December 2017”. Typo. “his wife his wife”.

Amended.

In fact, the purpose of this paragraph needs to be made clear. I am assuming that the point in outlining the 2003 and 2017 cases is to show/compare how the courts approach to penis taunting has change post reform? This needs to be made clear to the reader.

Amended.

 “A row” very British term…replace with an argument. “Sentence: lie with a 20-year minimum. He received the mandated life sentence with the minimum sentence set at 20 years”. Confusing, delete “lie with a 20-year minimum”.

Amended.

Again, referring to the Australian situation the author notes, “when leading the movement to reform the English law of murder, Solicitor General Harman was said to have favoured abolishing provocation outright, the preferred option in Australian jurisdictions”. Arguably, as per my previous comments, this is not the preferred option in Australia.

Amended.

The author then goes on to note that, “while conclusive findings must await more extensive comparative studies of pre-and post-reform cases across reform jurisdictions” (i.e. in Australia vs the UK). I would like to direct the author to the work of Kate Fitz-Gibbon in Australia who has been doing these types of analysis see https://www.researchgate.net/profile/Kate_Fitz-Gibbon. Any discussion regarding the Australian context should be made in references to Kate’s work as she is the leading authority on this subject. Overall, I would argue that the author should engage more directly with the Australian literature in the area and draw out points of comparison. This would strength the paper significantly.

Amended deleting the reference to comparative studies in order to keep the focus firmly on the English reforms.

Reviewer 3 Report

This is a really interesting topic and the central thesis is significant: That law reform has actually made a difference in terms of how the law is applied and the narratives told in the law and legal process. The use of Smart's work on feminist engagement with law is an interesting way of leading into the discussion. It also indicates the contribution that the article may potentially make to a broader and important issue - the degree to which engaging with the law can drive important reforms in relation to issues of structural injustice - as well as the very interesting more specific issue it directly tackles.

I did wonder how much of the change described in the article can be attributed to reform of the defences and how much can be attributed to other shifts in cultural narratives, including the work, for example, done by death review committees, along with other related law reforms, etc during the relevant time. This contextual background and some of the broader possibilities driving the shift in narratives are not outlined or explored. The assumption made throughout the paper - which is that the changes described are solely traceable to law reform relating to the defences to homicide - does seem a little simplistic.

The article needs a higher level of analysis to be really successful. At present the reader becomes bogged down in the factual detail of particular cases offered one after the other in a dense list of details, as illustrative of the general claims being made. There is some attempt to provide quantitative data but it is fairly general and the qualitative analysis is fairly factual and descriptive. It would be great to spend a bit more thinking time teasing out themes in the changes claimed - or indications of the changes claimed - with percentages indicating how frequently these occurred in the body of data. At the moment there are lots of references to "others" and "some".... How many of the 240 charged with murder plead guilty and how many went to trial? On what basis did they go to trial? How many were successful?

There is no attempt to compare case outcomes and narratives at either a quantitative or qualitative level with those prior to the reforms so the reader is asked to take on faith the significant changes claimed by the author. My own impression is that provocation, for example, was often raised in these cases pre the reforms but was increasingly unsuccessful. What was often offensive was the fact that the victim's family was put through days of character assassination of the victim - not that the defence was necessarily successful. That may not have been true in England (as the author implies) but the reader is simply asked to take this on faith. That seems to me to be a major methodological flaw in the argument.

The article needs an edit. At the moment there are a regular smattering of typos, non-grammatical sentences and colloquial English.

The author states at the beginning that "abolition of the provocation defence has been the reform favoured in several Australian jurisdictions," contrasting that with the English approach of reforming the offence. In fact one of the most important Australian states - NSW - also opted to reform their provocation defence rather than abolish it. It is not clear to me from the article why the English opted for reform if the main driver for change was the intention to abolish provocation by infidelity as a defence to murder.

The structure needs improvement - I am not sure what the sections "The Laws Truth Today" and "A history of the present" add to what has already been said. However, it could be that, as a reader I became a bit lost in the case detail and failed to grasp that there were new points being made.

This has the potential to be a really great article that - at present - reads as a bit of a first draft - one that requires more analysis, perhaps more research and a rewrite. It could be a great article as opposed to being a bit average and, in parts, a little sloppy.

Author Response

Review 3

This is a really interesting topic and the central thesis is significant: That law reform has actually made a difference in terms of how the law is applied and the narratives told in the law and legal process. The use of Smart's work on feminist engagement with law is an interesting way of leading into the discussion. It also indicates the contribution that the article may potentially make to a broader and important issue - the degree to which engaging with the law can drive important reforms in relation to issues of structural injustice - as well as the very interesting more specific issue it directly tackles.

I did wonder how much of the change described in the article can be attributed to reform of the defences and how much can be attributed to other shifts in cultural narratives, including the work, for example, done by death review committees, along with other related law reforms, etc during the relevant time. This contextual background and some of the broader possibilities driving the shift in narratives are not outlined or explored. The assumption made throughout the paper - which is that the changes described are solely traceable to law reform relating to the defences to homicide - does seem a little simplistic.

Response:

No such assumption is made. The paper is concerned solely with comparing the success of defence narratives after the reforms. My research reveals that with allegations of infidelity by provocation no longer able to found a defence, defendants are now pleading guilty to murder or being found guilty of murder. 

The article needs a higher level of analysis to be really successful. At present the reader becomes bogged down in the factual detail of particular cases offered one after the other in a dense list of details, as illustrative of the general claims being made.

Response:

This article is an argument for the continuing purchase of a method that reads law as a site for contesting law’s truth, using English cases and reforms as a case study. To make that argument, ‘factual detail’ is essential. The method looks at how killing and excuses for it are ‘put into discourse’.

It is unclear what ‘higher level of analysis’ is required in a paper making the case for a particular methodology which has proved invaluable for feminist researchers.

There is some attempt to provide quantitative data but it is fairly general and the qualitative analysis is fairly factual and descriptive. It would be great to spend a bit more thinking time teasing out themes in the changes claimed - or indications of the changes claimed - with percentages indicating how frequently these occurred in the body of data.

Response:

This not a quantitative analysis requiring data analysis. It is discourse analysis. The ‘themes’ emerging from the case law are highlighted from the changing discourse.

At the moment there are lots of references to "others" and "some".... How many of the 240 charged with murder plead guilty and how many went to trial? On what basis did they go to trial? How many were successful?

Response:

The reviewer must have missed this section.

Of the 240 (75 per cent of those charged) who pleaded guilty to murder or were convicted of murder by a jury convicted of murder, three received a whole life sentence, including two who had killed previous partners. The rest were given life sentences with minimum terms ranging from 11 to 38 years, far longer than sentences handed down prior to the reform. Ninety defendants admitted murder from the start or did so when it became clear during the trial that they had no viable defence.

….The study shows that the chances of a man avoiding a murder conviction for killing a woman partner or former partner have diminished substantially. Of the 36 defendants running the new loss of control defence, 27 were found guilty of murder.

There is no attempt to compare case outcomes and narratives at either a quantitative or qualitative level with those prior to the reforms so the reader is asked to take on faith the significant changes claimed by the author.

Response:

This is inaccurate. The paper discusses the controversial Humes case which precipitated the reforms and directs readers to my other published work discussing other pre-reform cases. This is not a matter of taking my claims ‘on faith’. It is a matter of public record that risibly short sentences for manslaughter handed down to wife-killers before the reforms, and the Humes cases specifically, precipitated the reform movement.

 My own impression is that provocation, for example, was often raised in these cases pre the reforms but was increasingly unsuccessful. What was often offensive was the fact that the victim's family was put through days of character assassination of the victim - not that the defence was necessarily successful. That may not have been true in England (as the author implies) but the reader is simply asked to take this on faith. That seems to me to be a major methodological flaw in the argument.

Response:

Here the reviewer relies on his/her ‘own impression’. My paper is based on the actual case law, not impressions.

The article needs an edit. At the moment there are a regular smattering of typos, non-grammatical sentences and colloquial English.

Response: I have amended the typos and grammar errors pointed out by the other 2 reviewers.

The author states at the beginning that "abolition of the provocation defence has been the reform favoured in several Australian jurisdictions," contrasting that with the English approach of reforming the offence. In fact one of the most important Australian states - NSW - also opted to reform their provocation defence rather than abolish it. It is not clear to me from the article why the English opted for reform if the main driver for change was the intention to abolish provocation by infidelity as a defence to murder.

Response:

I have amended the reference to the Australian reforms to make it clear that they are not the focus of this article.

Why the English reformers opted for reform is not relevant to my paper.

The structure needs improvement - I am not sure what the sections "The Laws Truth Today" and "A history of the present" add to what has already been said. However, it could be that, as a reader I became a bit lost in the case detail and failed to grasp that there were new points being made.

Response

There is no section called ‘The law’s truth’. It is ‘Law’s truth’, a reference to Carol Smart’s approach to the law question which traces shifts in law’s discursive construction. Without a passing familiarity with this methodology one could indeed become lost in the case detail as this reviewer suspects he/she has.

Round 2

Reviewer 3 Report

The manuscript has not been revised as I suggested so my review recommendations remain the same.

Author Response

Response 2 to Reviewer 3:

I am disappointed that this reviewer has not acknowledged the responses I made to his/her queries.

 The GE has reviewed all the reports and the revised manuscript and believes
that it is appropriate to proceed to publication of the article as it now stands. 
She states further that the author should not be required to write a completely different essay.